# Can pollen affect precipitation?

Marje Prank[1], Juha Tonttila[2,3], Xiaoxia Shang[2], Sami Romakkaniemi[2], Tomi Raatikainen[1]

[1]Climate System Research Unit, Finnish Meteorological Institute, Helsinki, 00560, Finland
[2]Atmospheric Research Centre of Eastern Finland, Finnish Meteorological Institute, Kuopio, 70211, Finland
[3]CSC - IT Center for Science Ltd., Espoo, 02101, Finland

*Correspondence to*: Marje Prank (marje.prank@fmi.fi)

**Abstract.** Large primary bioparticles such as pollen can be abundant in the atmosphere, for example near surface pollen concentrations above 10 000 particles per cubic meter can occur during intense pollination periods. On one hand, due to their

large size (10-100 micrometres), pollens can act as giant cloud condensation nuclei and enhance the collision-coalescence process in clouds that leads to drizzle formation. On the other hand, in humid conditions pollens are known to rupture and release many fine particles that can increase the cloud stability by reducing the droplet size. Additionally, both whole pollen grains and the sub-pollen particles released by pollen rupture are known to act as ice-nucleating particles (INPs). Due to these complex interactions, the role of pollen in modulating the cloud cover and precipitation remains uncertain.

We used the UCLALES-SALSA large eddy simulator for simulating birch pollen effects on liquid and mixed-phase clouds. Our simulations show that the pollen concentrations observed during the most intense pollination seasons can locally enhance precipitation from both liquid and mixed phase clouds, while more commonly encountered pollen concentrations are unlikely to cause a noticeable change. The liquid precipitation enhancement depended linearly on the emitted pollen flux in both liquid and mixed phase clouds, however, the slope of this relationship was case dependent. Ice nucleation happened at relevant degree

only if the process of rupturing pollens producing large number of fine ice nucleating particles was included in the simulations. The resulting precipitation saturated for the highest INP concentrations. Secondary ice formation by rime splintering had only minor effect in the considered one-day timescale.

# 1 Introduction

Large primary biological aerosol particles (PBAPs) such as pollen can be abundant in the atmosphere. For instance, according to Ranta and Satri (2007), the daily mean concentrations of birch plus alder pollen in Finland can exceed 1 000 pollen/m3 for up to 2 weeks per year. Daily mean concentrations above 10 000 pollen/m3 occur during intense pollination periods - in United States such pollen concentrations from both deciduous and evergreen trees have been observed (Steiner et al., 2015) and bihourly concentrations exceeding 50 000 birch pollen/m$^3$ were observed in Vehmasmäki station in Finland in May 2021. In

the current study we concentrate mostly on birch pollen as one of the most abundant pollens in boreal and northern temperate climates of Northern Hemisphere, such as Central and Northern Europe (Skjøth et al., 2013) or North America (Steiner et al., 2015). Birch pollen is also well studied due to its high allergenicity.

The abundance of pollen in many tree species including birch and alder varies between years depending on weather conditions and flowering intensity of previous year (Dahl et al., 2013). Pollen concentrations also exhibit regular diurnal variations with

afternoon peaks, although Rantio-Lehtimäki et al. (1991) found the concentrations of tree pollen to stay relatively constant over the day, with slight minimum in early morning.

Typically cloud droplets are formed on hygroscopic aerosol particles (composing of sulfate, nitrate, sea salt, organic aerosol, etc.), with the number of cloud condensation nuclei (CCN) in cubic centimeter ranging from below 100 in clean marine atmosphere to thousands in polluted areas (Seinfeld and Pandis, 1998). Compared to these numbers, even the highest observed

pollen concentrations are too low to noticeably influence the CCN concentration. However, in humid conditions pollens are known to rupture and release a large number of fine sub-pollen particles (SPPs) (Aznar et al., 2024; Emmerson et al., 2021; Stone et al., 2021; Suphioglu et al., 1992; Taylor et al., 2004; Wozniak et al., 2018). These SPPs acting as extra CCN can increase the cloud stability by reducing the droplet size.

On the other hand, as shown by Houghton (1938), rain drops in liquid clouds are formed by collisions of cloud droplets of

different sizes, and presence of at least a small number of extra-large hygroscopic CCN is an essential factor for the appearance of cloud droplets of different size in the same location. Coarse aerosols such as sea spray and mineral dust have been shown to act as such giant CCN (GCCN), enhancing the collision-coalescence process that leads to drizzle formation (Adebiyi et al., 2023; Feingold et al., 1999). Due to their large size (10-100 micrometres in diameter), pollens can also act as GCCN. Feingold et al. (1999) found noticeable enhancement in drizzle formation from GCCN concentrations as low as 0.001 cm$^{-3}$, which is

well in the range of observed pollen concentrations.

At temperatures warmer than the homogeneous freezing limit at about -38 °C, ice in the clouds is formed heterogeneously on particles which can initiate freezing. Depending on temperature, different solid particles can act as ice nucleating particles (INP). In colder temperatures (below -15 °C) ice nucleation is dominated by mineral dust, while primary biological aerosol particles are the most efficient INPs for temperatures warmer than -10 °C (Hoose and Möhler, 2012). Both pollen and SPPs

can act as INPs (Pummer et al., 2012). Pollens of spring flowering trees such as birch and alder have been shown to be good ice nucleators in relatively higher temperatures (Dreischmeier et al., 2017; Gute and Abbatt, 2020).

The processes leading to pollen rupture are not well understood. While sub-pollen particles from birch pollen have been observed both in lab and in atmosphere (Burkart et al., 2021; Rantio-Lehtimaki et al., 1994; Schäppi et al., 1997), the frequency of this process happening in atmosphere has not been quantified. More research exists about grass pollen related to asthma outbreaks coinciding with thunderstorms, but conditions leading to it have not been well quantified (Emmerson et al., 2021). The only data available to our knowledge regarding pollen rupture due to high air humidity was collected by Zhou (2014) for wheat and pine pollens. Wheat pollen was the only one rupturing in their setup. Unfortunately, they made no experiments with birch pollen. Large uncertainties exist also in the number of SPPs released from a rupturing birch pollen and their size distribution.

Large spread exists also in the measurements of ice nucleation efficiency by pollen and SPPs. The median freezing temperatures measured for birch pollen reach from -13.4 to -27 °C and for alder from -7.3 to -17 °C and are sensitive to atmospheric processing experienced by the pollen grains (Gute et al., 2020; Gute and Abbatt, 2020). The temperature of freezing onset is challenging to measure and uncertain (Duan et al., 2023), however, Wieland et al. (2024) showed that birch pollen can nucleate ice in temperatures at least up to -5.4 °C.

Small number of global and regional modelling studies have investigated the impact of pollen and SPPs to precipitation, however, often the pollen concentrations in those are low, representative of long-time or large-scale averages, (e.g. Werchner et al., 2022; Wozniak et al., 2018). While some studies (e.g. Zhang et al., 2024) have used realistic pollen emissions, their emissions represent an average pollen year, while during intense flowering on mast years the concentrations can locally reach many times what is used in those studies. Also, while ice nucleation is included in some of the studies, the precipitation parameterizations in global and continental scale models do not explicitly account for the GCCN effects of large particles.

In this study we apply UCLALES-SALSA large eddy simulator (Tonttila et al., 2017, 2021) to explore what kind of birch pollen concentrations are required to impact precipitation in liquid and mixed phase clouds in local scale. We quantify the CCN, GCCN and INP effects of pollen and SPPs for a range of pollen concentrations and test the effect of different assumptions about SPP size distribution. The simulations allow us to quantify the fraction of pollen and SPPs that escape the boundary layer to free troposphere and can participate in long-range transport in different cloud conditions.

## 2 Methods

### 2.1 Model description

We used the UCLALES-SALSA model that combines the UCLA Large Eddy Simulator (Stevens et al., 1999, 2005; Stevens and Seifert, 2008) with the Sectional Aerosol module for Large Scale Applications (SALSA , Kokkola et al., 2008, 2018) and includes representations of aerosols, liquid cloud droplets, raindrops and ice and simulates their interactions (Ahola et al., 2020; Tonttila et al., 2017, 2021). The version of UCLALES-SALSA used in this study explicitly computes rain drop formation through collision-coalescence of cloud droplets. This process is included in the collision scheme that handles all collisions between different types of particles including coagulation of aerosol particles, coalescence of cloud droplets, accretion of liquid

droplets to ice particles and scavenging of aerosol and cloud droplets by falling raindrops or ice. The cloud droplets are moved to rain phase when their wet diameter after collision exceeds minimum drop size of 20 µm. When liquid droplets leave the cloud to subsaturated conditions and enough water evaporates to bring the particles close to equilibrium with the ambient relative humidity, they are moved back to aerosol phase.

The model was amended with parameterizations for pollen emission and humidity dependent rupture. Pollen is emitted as constant flux from surface. We simulate pollens as spherical particles, birch pollen with diameter of 22 µm and density of 800 kg/m$^3$ (Gregory, 1961) and pine pollen with diameter of 59 µm and density of 450 kg/m$^3$ (Jackson and Lyford, 1999). Direct emission of sub-pollen particles from trees is not considered in this study, as there is no data available to quantify such flux.

As no measurement data about humidity dependent pollen rupture rate is available for birch pollen, we followed the example of Werchner et al. (2022) and parameterized the rupture process as exponential decay with timescale and humidity dependence approximated from the data of Zhou, (2014) for wheat pollen. Pollens start rupturing when relative humidity exceeds 80% and their e-fold lifetime reduces linearly from 12.5 to 2.5 hours as humidity increases from 80% to 95%. We limit the e-fold lifetime to minimum half an hour. This exponential decay approximation would overestimate the rupture rate at the beginning of the Zhou's (2014) experiments where the temporal development is looking more sigmoidal, however, the slow starting rate in the lab could be due to using dry pollen, while in nature the pollens have been exposed to ambient humidity already in the catkins. Thus, given all the uncertainties, we selected to use the simplest form of parameterization. The mass of the rupturing pollens is reduced by the total mass of the released SPPs, after which they continue interacting with the clouds as GCCN and INP but are not allowed to rupture again. Pollens rupture much faster when fully immersed in water, as is the case when a pollen is included in a raindrop. However, it is unclear if the released fragments would be able to leave the raindrop or they would stay inside the drop and stick to the surface of the pollen grain when the drop evaporates. For simplicity, we simulate no SPP release from raindrops and consider the pollens that have been in raindrops incapable of rupturing further.

The size of the SPPs affects their ability to act as both CCN and INP. To explore the impact of different assumptions regarding the size of the SPPs, lognormal distributions (Figure 1) were fitted to the measurements from Taylor et al., (2004) (mean diameter 0.3 µm, geometric standard deviation 2.2, referred as T04 further on) and Burkart et al., (2021) (mean diameter 1.15 µm, geometric standard deviation 1.5, further referred as B21). Following Wozniak et al. (2018), we assumed 1000 sub-pollen particles to be emitted from each ruptured pollen, which is similar to assessments of Stone et al. (2021) and Suphioglu et al. (1992).

In addition to size and density, the cloud interactions are also sensitive to particle's hygroscopicity. As a simplification, we treat pollen as a soluble particle and set the same hygroscopicity parameter to 0.16 for both SPPs and whole pollens, which is consistent with the critical supersaturation measurements for birch pollen SPPs by Steiner et al. (2015). As in reality a whole pollen is not a soluble particle, use of kappa-Koehler theory (Petters and Kreidenweis, 2007) is not exactly correct for it. However, as particles with diameters in the range of tens of micrometres activate easily as cloud droplets as long as they are not hydrophobic, this approximation should have limited impact. Griffiths et al. (2012) reported pollen hygroscopicities in the

range of 0.05 to 0.22 and found that the wettability and large size of pollen grains leads to them activating to cloud droplets in supersaturations of 0.0015% and lower.

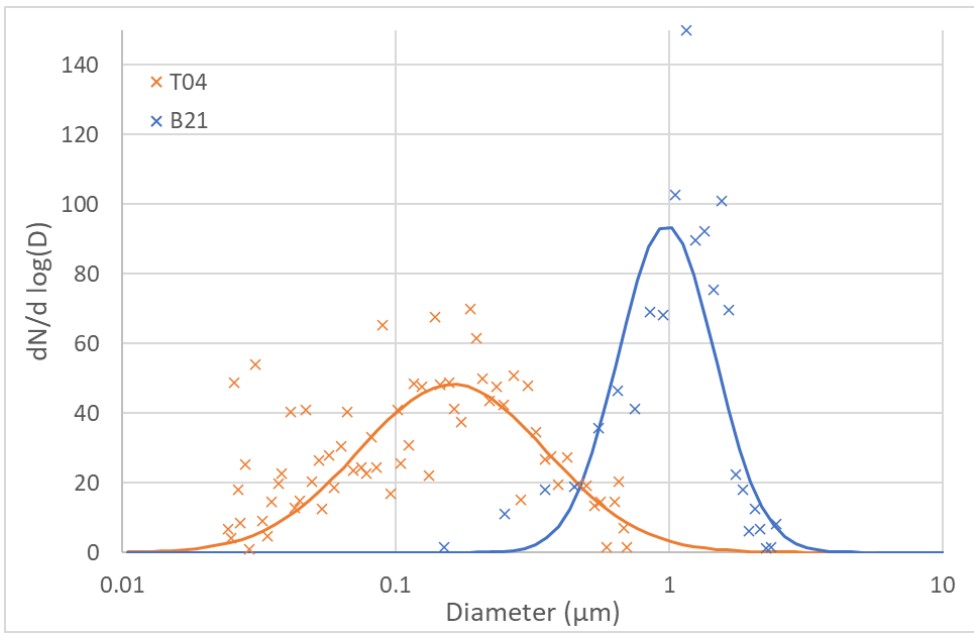

**Figure 1: Size distributions fitted to data from Taylor et al., (2004) (orange) or Burkart et al., (2021) (blue).**

Ice nucleation parameterization for both pollen and sub-pollen particles was based on a simplified version of the Augustin et al. (2013) scheme. The temperature dependence of the ice nucleation rate is computed as an exponential fit: $j = 2.32 \times 10^{-8} \times e^{-0.835 \times T_c}$ where $j$ is the heterogeneous nucleation rate per second and $T_c$ is the temperature in Celsius. Augustin et al. (2013) also parameterized the average number of ice-nucleation active macromolecules in SPPs depending on their diameter and we use this parameterization to account for the probability that a small SPP particle might not contain any ice-nucleation-active macromolecules. However, to avoid unrealistically high ice nucleation rates for whole pollen grains and larger SPPs in near-zero temperatures, we assume that there is exactly one ice nucleation active site in every ice-nucleation active particle (SPP, or whole or ruptured pollen). Augustin et al. (2013) showed in their Appendix A that this approximation reproduces well the slope of the frozen fraction for SPPs. For whole pollens it gives us a median freezing temperature of ~ -15 °C at a cooling rate of 0.67 K per minute used by Gute and Abbatt (2020), which is slightly lower than their measured median freezing temperature -13.4 °C but on the higher end of the rest of their reviewed data. It also agrees well with the model of Hoose et al. (2010), who report the temperature below which the freezing rate exceeds $10^{-5}$ s$^{-1}$ for birch pollen at approximately −8 °C, while our model reaches this rate at -7.24 °C. The freezing rate in our model is zeroed for temperatures above -2 °C. All particles formed by collisions with ice are assumed to freeze. Secondary ice formation through rime splintering (Hallett and Mossop, 1974) is included in the simulations. Splinters are formed at temperatures between -3 and -8 °C with

$3.5 \times 10^8$ splinters produced per kilogram of rime at the optimal -5 °C temperature. The parameterization of Seifert et al. (2014) for cloud ice is used for the effective size and terminal velocity of the ice particles.

## 2.2 Model simulations

To investigate the impact of high pollen concentrations to cloud processes, we simulate two well described cases – one for liquid and one for mixed-phase clouds.

For liquid clouds we use the Rain in Cumulus over the Ocean (RICO) Field Campaign characterized by lightly precipitating cumulus-topped boundary layer, adapted for large eddy simulator (LES) studies by VanZanten et al. (2011), who selected the case as a simple prototype for precipitating convective clouds. The field campaign took place over the Northwestern Atlantic in winter. The clouds in this case are shallow enough to simulate with LES without the domain size becoming computationally prohibitively expensive while still being deep enough for precipitation development, and sensitive to microphysics including aerosol perturbations. The case allows us to investigate the role of convective structures in transporting particles such as pollen and SPPs from boundary layer to the free troposphere where they can be transported for long distances and contribute to cloud and ice nucleation processes in larger scale. The near surface temperature during the campaign was +26 °C. Such high temperature on a spring day would result in rapid pollen maturing and release and thus would lead to very high pollen concentrations. The temperature was above zero throughout the cloud layer.

For mixed phase clouds we use the second case described by (Calderón et al., 2022). The measurement campaign that their LES simulations were based on took place in Puijo station, Finland. The case is characterized by a low-level stratocumulus cloud with low aerosol loading and light drizzle formation in cloud. The near-surface temperature was about 0°C and some ice particles were observed. We extended the simulations to 24 hours from the original 6 hours to allow the pollen to be transported to the cloud, rupture to produce SPPs, and ice nucleation to take place. The cloud top temperature at the beginning of the episode was about -3 °C and was falling while the cloud top was rising. These temperatures are ideal for secondary ice production through rime splintering (Hallett and Mossop, 1974). While high pollen emission is unlikely with near-zero temperature, Puijo station is elevated compared to its surroundings where the temperature is likely to be higher. Also, the case starts at midnight, so the pollen could have matured during the warmer afternoon while its release from catkins could have been delayed due to too high humidity or low wind conditions. The case also gives us a chance to investigate if any pollen or SPPs manages to escape the boundary layer through the relatively strong inversion that was present in this case.

In both cases preliminary model simulations were used to calibrate the birch pollen emission flux to produce near surface concentrations covering from commonly observed daily mean values of about 1000 up to the hourly maximum values of more than 50000 pollens per $m^3$. Fluxes required in the simulations to produce these concentrations are shown in Table 1. Simulations with T04 SPP distribution were made only for the maximum pollen flux to save computational resources. To investigate the impact of pollen size to the GCCN effect, one extra simulation was made for the RICO case in which the pollen emission flux was kept the same as in the maximum birch pollen flux but pine pollen size and density were assigned. Pine pollen can be present in atmosphere in large quantities. Its diameter is about 3 times larger than that of birch pollen however,

pine pollens incorporate two large air bladders that reduce its density and help with buoyancy. As Zhou, (2014) found no rupture for pine pollen, we do not consider this process for the pine pollen simulation.

The setup of the simulations is shown in Table 1. Both cases were run for 24 hours with 1 second internal timestep, which is further shortened if needed for model stability. Model spin-up allows the turbulence to develop while precipitation formation is turned off. All simulations are initialized with background aerosol size distribution and properties following the specifications in the case publications (Table 1). Table 2 and Table 3 list all the simulations with their acronyms and parameters that were varied for each of the cases.

**Table 1. Model setup**

| Cloud type | Liquid cumulus | Mixed-phase |
|---|---|---|
| Case reference | RICO, VanZanten et al. (2011) | Puijo, Calderón et al. (2022) |
| Domain size | 12.8 x 12.8 km | 1.92 x1.92 km |
| Horizontal resolution | 80 m | 30 m |
| Domain height | 4 km | 1.2 km |
| Vertical resolution | 30m | 10 m |
| Simulation length | 24 h | 24 h |
| Internal timestep | 1 s | 1 s |
| Output timestep | 3 min | 1 min |
| Spinup | 2 h | 1 h |
| Background aerosol<br>Number of lognormal modes:<br>geometric mean diameters:<br>geometric standard deviations:<br>concentrations: | Ammonium-bisulfate<br>2<br>0.06 and 0.28 µm<br>1.28 and 1.75<br>90 and 15 $cm^{-3}$ | 12% sulfate, 88% organic carbon<br>3<br>0.039, 0.215 and 0.735 µm<br>1.5249, 1.5826 and 1.1811<br>274, 93 and 15 $cm^{-3}$ |
| Pollen emission calibration: surface flux and resulting near-surface concentration | P30: 30 #/$m^2$/s -> 1340 #/$m^3$<br>P300: 300 #/$m^2$/s -> 13000 #/$m^3$<br>P1500: 1500 #/$m^2$/s -> 60500 #/$m^3$ | P50: 50 #/$m^2$/s -> 1180 #/$m^3$<br>P500: 500 #/$m^2$/s -> 11500 #/$m^3$<br>P2500: 2500 #/$m^2$/s -> 56200 #/$m^3$ |

**Table 2. Model simulations for the liquid cumulus case (RICO) with all the varied parameters**

| Simulation acronym | Pollen emission flux (#/m$^2$/s) | Sub-pollen particle size distribution | Pollen dry diameter (µm) | Pollen dry density (kg/m$^3$) |
|---|---|---|---|---|
| No ems | 0 | No rupture | 22 | 800 |
| P30 no-SPP | 30 | No rupture | 22 | 800 |
| P30 SPP-B21 | 30 | B21 | 22 | 800 |
| P300 no-SPP | 300 | No rupture | 22 | 800 |
| P300 SPP-B21 | 300 | B21 | 22 | 800 |
| P1500 no-SPP | 1500 | No rupture | 22 | 800 |
| P1500 SPP-B21 | 1500 | B21 | 22 | 800 |
| P1500 SPP-T04 | 1500 | T04 | 22 | 800 |
| P1500 pine | 1500 | No rupture | 59 | 450 |

**Table 3. Model simulations for the mixed-phase cloud case (Puijo) with all the varied parameters**

| Simulation acronym | Pollen emission flux (#/m$^2$/s) | Sub-pollen particle size distribution | Secondary ice formation |
|---|---|---|---|
| No ems | 0 | No rupture | No |
| P50 no-SPP | 50 | No rupture | No |
| P50 SPP-B21 | 50 | B21 | No |
| P500 no-SPP | 500 | No rupture | No |
| P500 SPP-B21 | 500 | B21 | No |
| P2500 no-SPP | 2500 | No rupture | No |
| P2500 SPP-B21 | 2500 | B21 | No |
| P2500 SPP-T04 | 2500 | T04 | No |
| P2500 no-SPP SIP | 2500 | No rupture | Yes |
| P2500 SPP-B21 SIP | 2500 | B21 | Yes |
| P2500 SPP-T04 SIP | 2500 | T04 | Yes |

190

## 3 Results

### 3.1 Liquid cumulus case

To investigate the impact of pollen and SPPs on liquid clouds we simulated the RICO Field Campaign characterized by cumulus-topped boundary layer, adapted for large eddy simulator studies by VanZanten et al. (2011). Figure 2 shows the temporal evolution of selected variables during the simulations. The lines have been smoothed using singular spectrum analysis with 6-hour window length to make them easier to distinguish from each other and hourly averaged model output is plotted as dots to visualize the model variability. Comparing the grey no-emission timeseries with the coloured ones shows that in this case the pollen and SPPs do not affect the cloud dynamics in larger scale, as the difference in cloud height and maximum updraft velocity (Figure 2d, f) between the simulations does not exceed the model noise and the changes in cloud cover fraction on panel c (with the exception of the pine pollen simulation) are also minor. As seen from Figures 2h, 3b and 4a, the UCLALES-SALSA simulation of this case without pollen emission produces almost no precipitation within 24 hours. However, the GCCN effect of the pollens enhances the collision-coalescence rate in the simulations and leads to increased surface precipitation (Figure 2g, h). The domain averaged accumulated precipitation is low even for the maximum emitted pollen flux; however, some isolated larger clouds can produce noticeable precipitation rates (Figure 3a). As the clouds are moving with the wind, the precipitation does not reach the surface directly below them, and in some cases the cloud that produced it has already mostly dissipated.

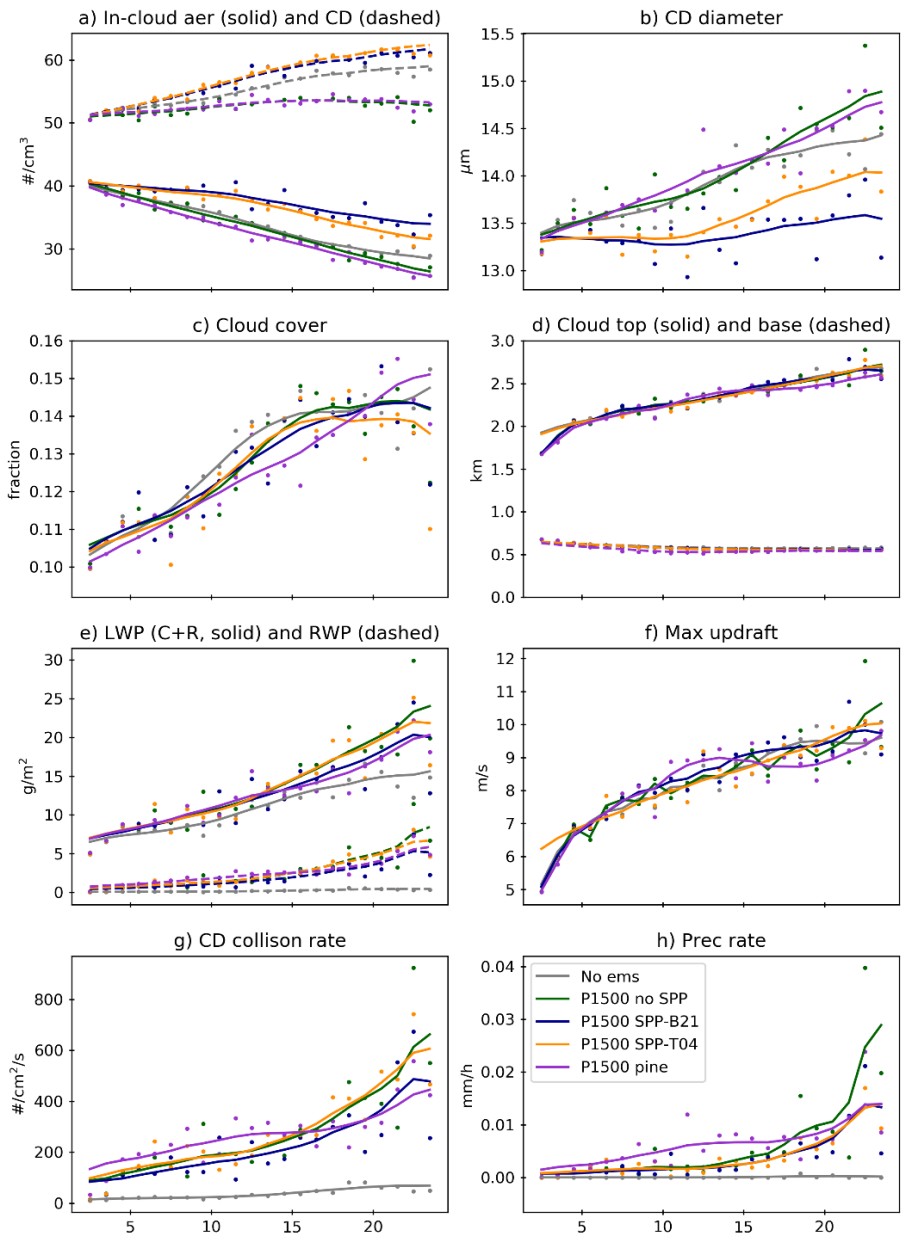

**Figure 2. Impact of pollen and SPPs on the clouds. Panels: a – in-cloud interstitial aerosol and cloud droplet concentration, b –cloud droplet size, mean over cloudy grid cells, c –cloud cover fraction, d – height of cloud top and base, e - liquid water path (cloud + rain) and rain water path, f – maximum updraft velocity, g – loss rate of cloud droplets due to collision-coalescence, mean over cloudy columns, h - mean precipitation rate at surface. Simulations: grey – no-emission control, green –birch pollen flux (1500 pollen/m2/s), no rupture, dark blue – same birch pollen flux, SPP size from B21, orange– same birch pollen flux, SPP size from T04, purple – pine pollen flux (1500 pollen/m2/s), no rupture. Hourly averaged time series, mean over the model area. Grid cell is considered cloudy if cloud water mixing ratio exceeds 1.e-5 kg/kg. Dots – hourly average model values, lines – trend component from singular spectrum analysis with 6 hour window length.**

Panel b of Figure 3 shows the histograms of the instant precipitation rate in every grid-column every output timestep (3 minutes) of the second half of the simulations, when the pollen and SPPs had had time to start influencing the precipitation. As the cloud cover fraction stayed below 15% in all simulations, we can expect no rain in at least 85% of the domain. Indeed,

220  precipitation stayed below 0.001 mm/h in more than 90 % of the domain in all simulations. As seen from Panel b of Figure 3, the shapes of the rain rate distributions are very similar between all the simulations and increasing pollen emission increases the number of precipitating grid-cells in every rate interval. This indicates that the increase in accumulated precipitation is not due to heavier rainfall from a few clouds but due to larger fraction of the clouds precipitating.

The precipitation enhancement is nearly linear to the pollen emission (Figure 4b). Including the pollen rupture process

225  increases the cloud droplet number and reduces the cloud droplet size, stabilizing the clouds and decreasing the precipitation (Figure 2a, b, h). While the cloud droplets are slightly smaller for the B21 parameterization compared with the T04 one (Figure 2b), it does not seem to influence the total precipitation in the highest emission case (Figure 4a), for which both of the parameterizations were tested.

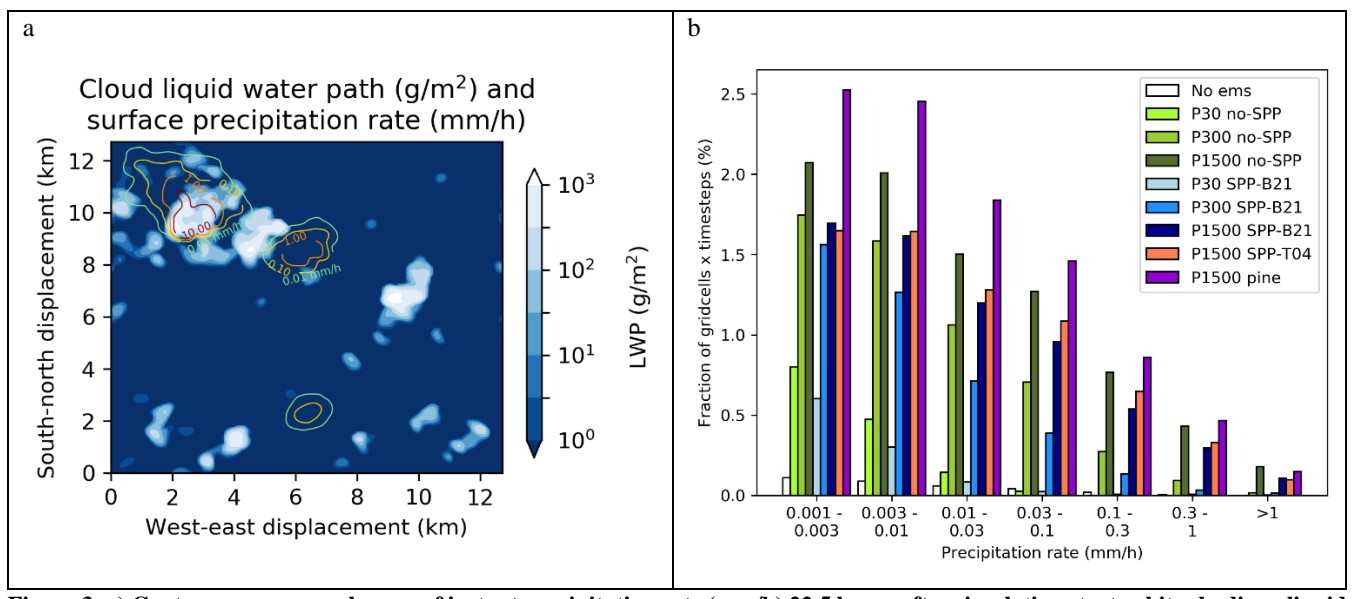

230  **Figure 3. a) Contours – an example map of instant precipitation rate (mm/h) 22.5 hours after simulation start, white shading –liquid water path (g/m²); b) histogram of instant grid cell rain rates in the second half of the simulation (mm/h). Precipitation rates below 0.001 mm/h are not shown.**

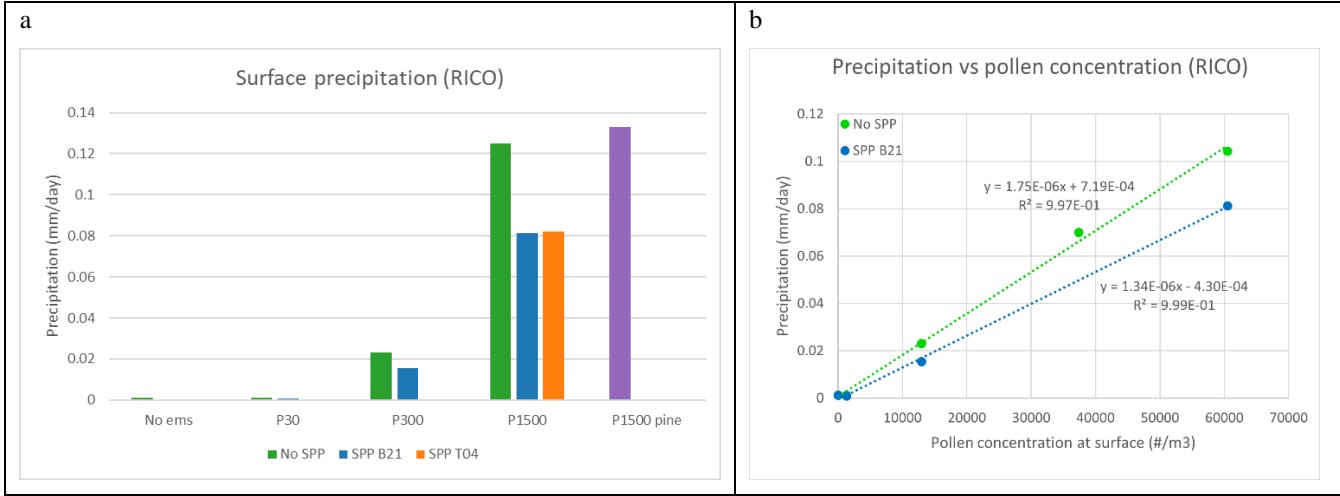

**Figure 4. A – total accumulated surface precipitation by the end of the 24 hour simulations of RICO case with various emissions. B - Accumulated precipitation vs mean pollen concentration near surface at the second half of the simulation; green – no rupture, blue (or orange) – SPP size distribution from B21 (or T04).**

Figure 5 and Figure 6 show the vertical profiles of pollen and SPP number concentration and its tendencies in aerosol, cloud and rain phases caused by various processes in the midpoint of the simulation with maximum pollen emission and rupture according to B21 size distribution. Pollen is emitted from the surface (Figure 5d, red line) and thus has a large vertical gradient near ground. Below cloud it is mostly in aerosol phase (Figure 5e). Cloud activation takes place at the cloud base (Figure 5a,b, green lines) and above that the pollen-containing cloud droplets quickly grow to raindrop size by coalescence (Figure 5c, blue line).

As seen from panel d of  Figure 6, the most intense pollen rupture takes place at the cloud base where the humidity is high and pollen concentration is still reasonably high. Pollen rupture is not visible on Figure 5 because the ruptured pollens are still tracked in the simulation, so their number does not change. From the cloud base the produced fine sub-pollen particles are mixed upwards and downwards (Figure 6d, pink line) and partly also immediately activated to cloud phase (green lines, Figure 6a,b). Cloud activation rate is much higher than rupture rate because pollen rupture in the model is a relatively slow process, with a time scale of 2.5 hours at highest humidities. Activation to cloud droplets at the cloud base, on the other hand, is a fast process that involves all the SPPs produced cumulatively during the simulation and is balanced by evaporation back to aerosol phase after leaving the cloud. Transport rate includes both grid scale and sub-grid scale vertical transport, and as domain average it can be considered as mixing that acts to reduce the vertical gradients produced by the other processes. The processes that move particles from one phase to the other are not visible on panel d of Figure 6 that depicts the sum of all the phases, so there the rupture followed with transport are the dominating processes. For SPPs sedimentation is too slow compared with other processes to be visible.

From Figure 6c we see that smaller particles such as SPPs form raindrops more uniformly throughout the cloud layer, while pollen including raindrops (Figure 5, panels c and e) form mostly close to the cloud base. Comparing the blue lines on Figure

5e and Figure 6e we see that the SPP containing raindrop concentration decreases much faster below the cloud base than the pollen containing raindrop concentration. The raindrops formed around pollen grains are larger and thus fall faster and have

260 less time to evaporate, leading to larger liquid water flux to surface, while almost all of the SPP containing raindrops evaporate already near the cloud base.

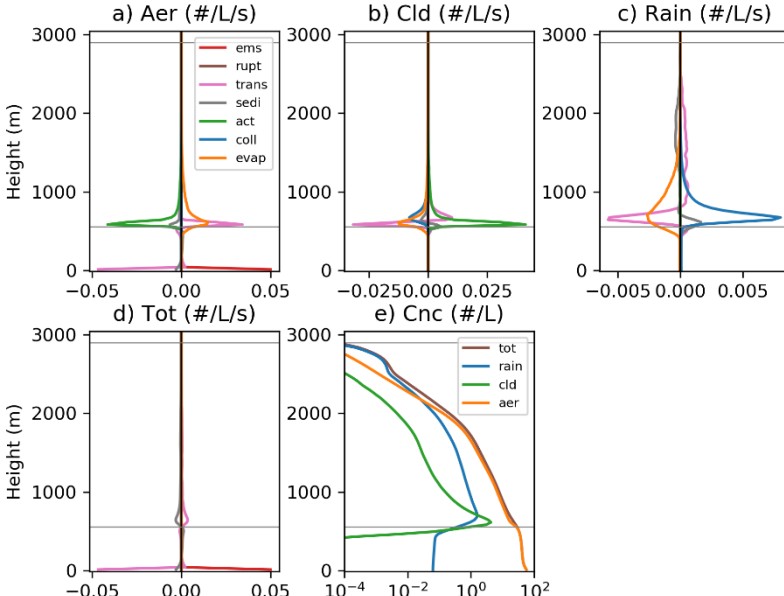

**Figure 5. Vertical profiles of pollen concentration tendencies in aerosol (a), cloud (b) and rain (c) phases, and the sum of those (d)**
265 **due to various processes: pollen emission, rupture, vertical transport, sedimentation, cloud activation, collisions and evaporation. E**
**– number concentration profiles of different phases. Domain averaged hourly mean quantities 12 hours after the beginning of the**
**simulations, plotted for the case with maximum pollen flux and rupture according to B21 size distribution. Grey lines denote the**
**cloud base and top of highest clouds. Both whole and ruptured pollen grains are included.**

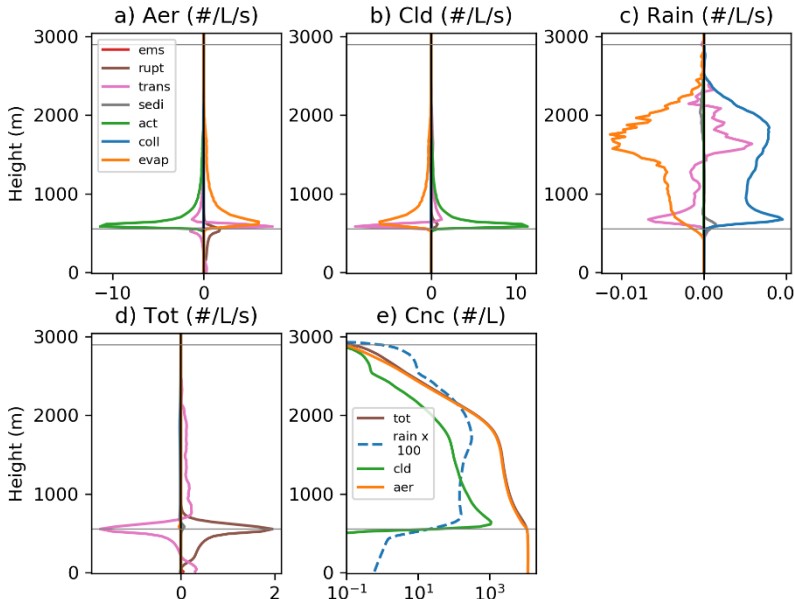

**Figure 6. A-D Vertical profiles of sub-pollen particle concentration tendencies in different phases due to various processes. Panels and colors same as in Figure 5. E – number concentration profiles of SPPs in different phases. Rain phase has been multiplied with factor of 100 to fit in the same scale.**

Pollen size has a two-fold effect on its ability to act as GCCN – on one hand large pollen would be more efficient in collecting cloud droplets to form precipitation but on the other hand the higher deposition rate of the larger particles would lead to much lower concentration at cloud level. For pine pollen the enhanced deposition resulted in three times lower near-ground concentration than for birch pollen and also faster drop off for higher vertical levels (Figure 7, B). However, the resulting precipitation started earlier (Figure 2,f) and was slightly larger than for the birch pollen case (purple bar on Figure 4).

Panel a of Figure 7 shows the normalized net flux of pine and birch pollens and birch SPPs through the inversion layer. The fluxes are positive for most of the simulation time, although for pine pollen the flux is very small. In the last hour of the simulation 0.03% of pine pollen, 0.14% of birch pollen and 1.86% of birch SPPs cumulatively emitted or produced during the whole simulation are located above the inversion layer. Majority of birch pollen and SPPs (88% and 98% respectively) above the inversion layer are found in aerosol phase, indicating that they have escaped the clouds through detrainment. As SPPs survive higher in the cloud than pollens, more of them are released from the evaporating cloud droplets at higher altitudes (Figures 5 and 6 ,panels a-c). By the end of the simulation, ~23% of the emitted pollens have ruptured, each releasing 1000 SPPs, leading to the number of SPPs above the inversion layer being 3 to 4 orders of magnitude higher than that of pollen.

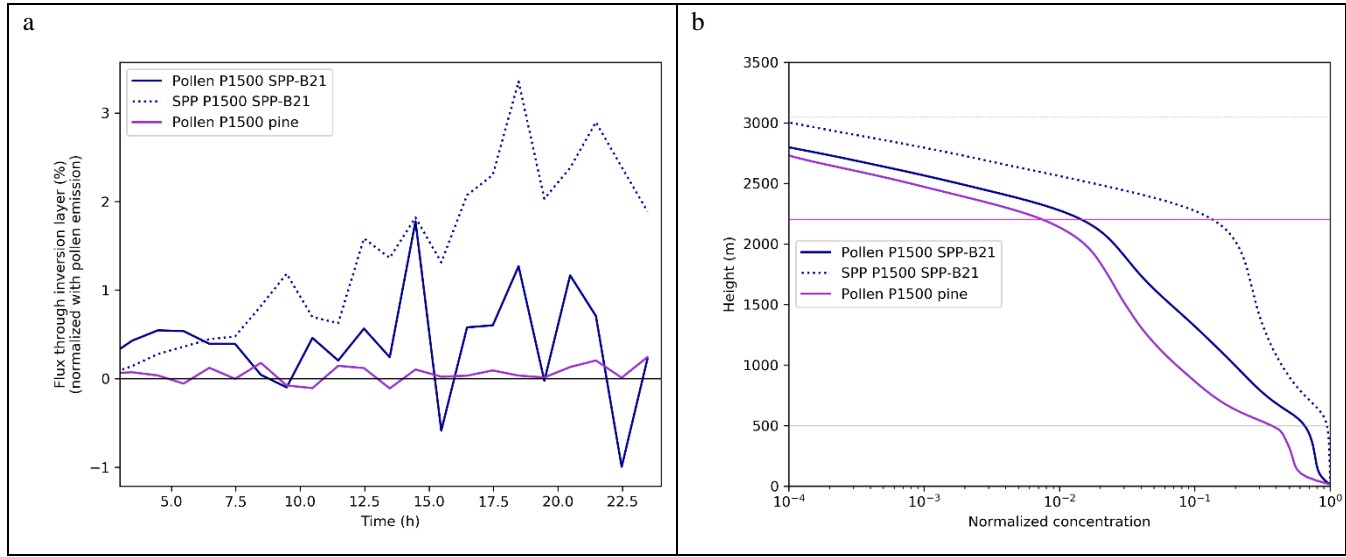

**Figure 7. A – Domain mean hourly averaged net flux of pollen and SPPs through the inversion layer. For pollen the flux is normalized with the pollen emission flux and for SPPs with pollen emission × 1000. B – Normalized domain average vertical profiles of pollen**
**concentration for birch (blue) and pine (purple) pollens and birch SPPs (blue, dashed) at the end of the simulation. The magenta line denotes the inversion layer, the solid gray the cloud base and the dashed gray the top of the highest clouds.**

## 3.2 Mixed phase case

For mixed phase clouds, we simulated the second case described by Calderón et al. (2022), a nocturnal low level stratocumulus
episode observed in Puijo, Finland. In the course of the 24-hour simulation the cloud top rises from below 400 m to above 800 m and the cloud top temperature falls from -3 to -7 °C (Figure 8). Some snowflakes were observed during the measurement campaign but UCLALES-SALSA simulation without pollen emission produces almost no precipitation within 24 hours and as no INPs were included in the background aerosol, no ice is formed.

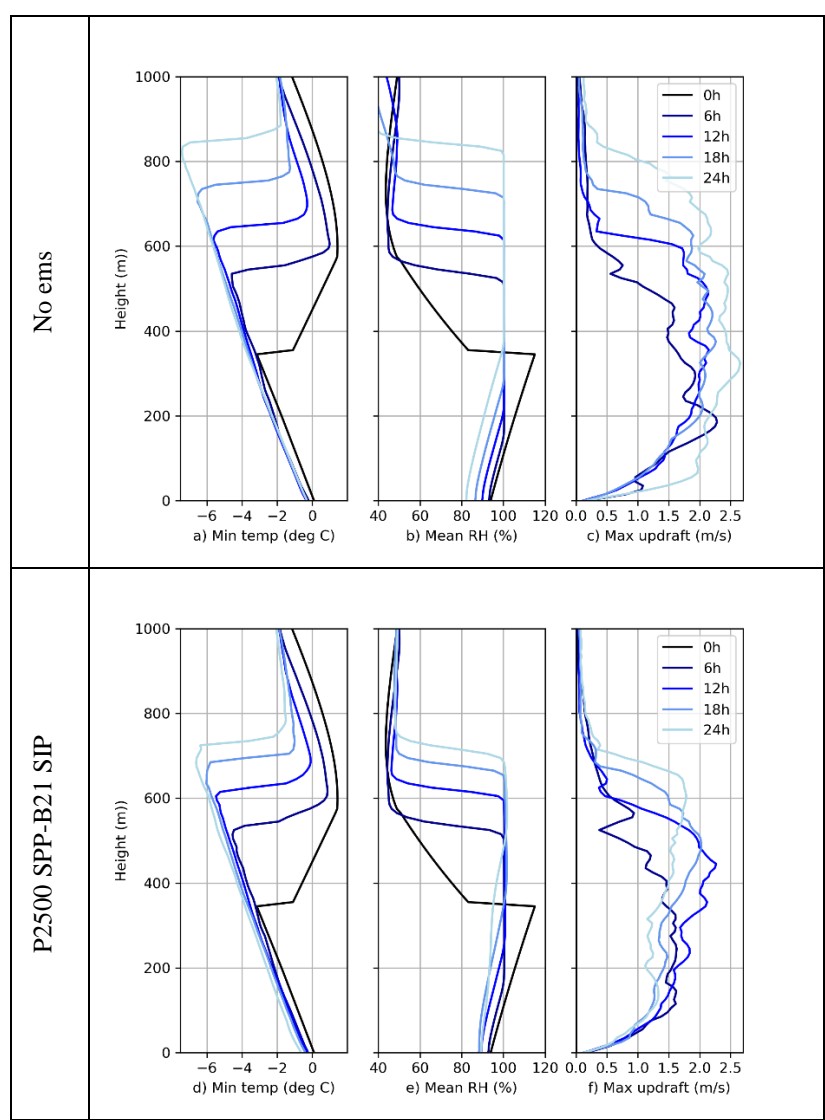

**Figure 8. Profiles of domain minimum temperature, mean relative humidity and maximum updraft velocity for the no-emission control (upper row) and the maximum INP simulation (P2500 SPP-B21 SIP, lower row) (initial, 6, 12, 18 and 24 hours after simulation start, from darker to lighter lines).**

As seen from comparing the grey and light green lines on Figure 9 (panels g and h), introduction of pollen to the simulations without the rupture process leads to light liquid phase drizzle due to the GCCN enhancing the collision rate, however, the pollen concentration in the clouds is too low to cause significant ice nucleation even for the maximum pollen flux (Figure 9c). The rupture of a pollen grain produces 1000 potentially ice nucleating SPPs that also have slower settling velocity and thus their concentration in the cloud top can be much larger. As the relative humidity in this case was above 80% from surface to

cloud top (Figure 8, right), the rupture rate of pollens was relatively fast. While these additional particles reduce the cloud droplet and ice particle size (Figure 9b), they also make the cloud start to glaciate. While even in the case with maximum INP production (P2500 SPP-B21) the ice particles make up only a small fraction (<0.02%) of the total cloud droplets (Figure 9c), their effects are well noticeable. Once formed, they grow fast and end up being much larger than the cloud droplets (Figure 9b), so by the end of the simulation with maximum ice formation, about half of the total (liquid + ice) water path is actually

frozen (Figure 9e). Due to the faster settling of the ice particles, the total water path also reduces by half and a drop is visible in the cloud height by the end of the simulation (Figure 9d). In fact, enough water is removed from the cloud layer by settling of the ice particles and subsequently sublimated back to vapour phase below cloud base to change the humidity profile. In the no-emission base case, the relative humidity at surface falls to ~80% by the end of the simulation (Figure 8b), while for the maximum ice nucleation case it stays around 90% and a small drop is also seen in the near-surface temperature due to the

energy spent for vaporization of the falling ice ((Figure 8d,e). This results in less buoyancy and slowdown of updraft velocities (Figure 8c,f, Figure 9f).

The additional INPs from pollen rupture lead to noticeable solid phase precipitation (Figure 11a), especially if assuming the larger size distribution (B21), as according to the used ice nucleation parameterization, not all the smallest SPPs include ice nucleating macromolecules. For total precipitation, the large impact of additional INPs dominates over the competing effect

of the extra CCN reducing cloud droplet size. As pollen rupture is a relatively slow process, it takes several hours for the SPP ice nucleation effect to start dominating over the pollen GCCN effect (Figure 9h). Secondary ice production through rime splintering has minimal effect to the simulations (Figure 11a), with the exception of the no-SPP case where the ice precipitation rate starts rising at the end of the simulation (Figure 9, panel h), indicating that the process could become important in longer timescale.

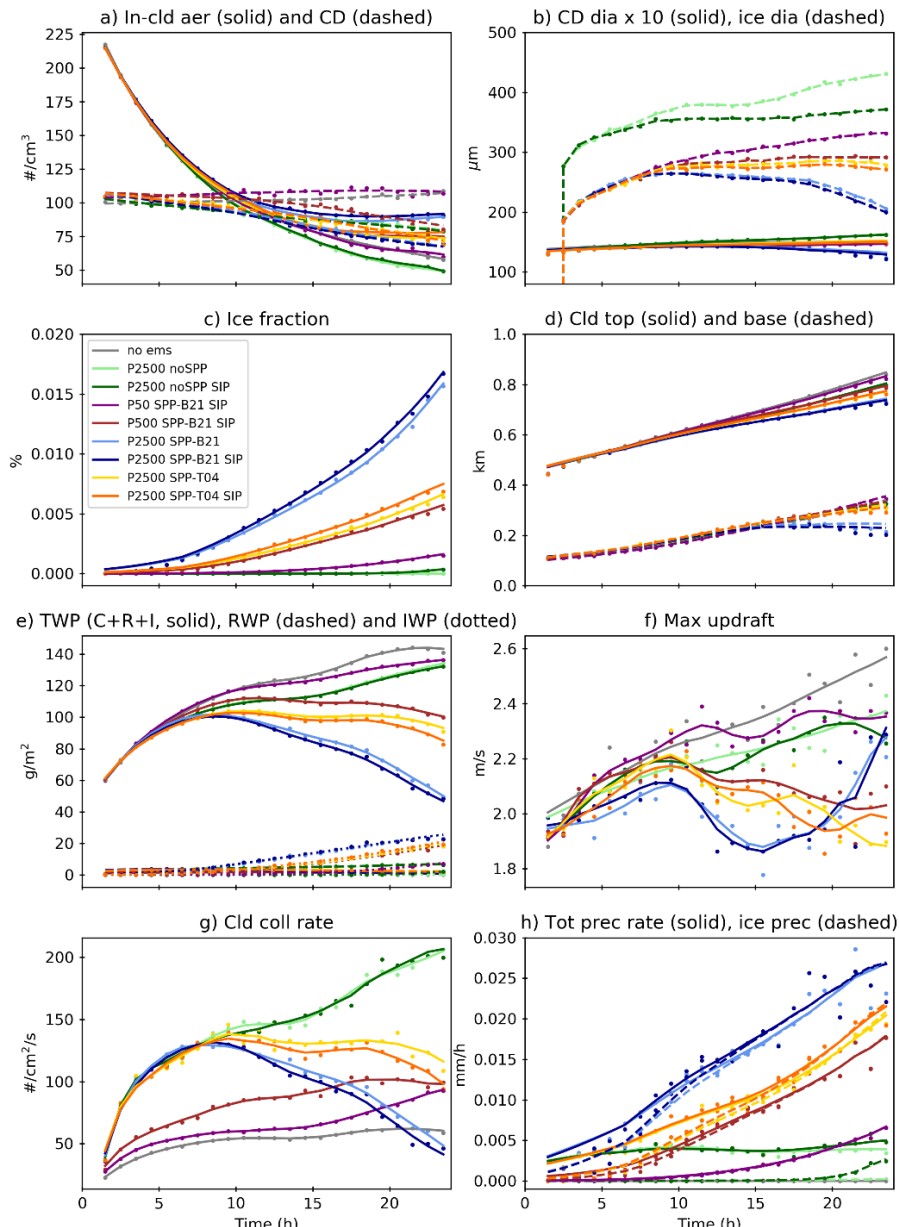


**Figure 9. Impact of pollen and SPPs on the clouds. Panels: Panels: a – in-cloud interstitial aerosol and cloud droplet concentration, b –cloud droplet and ice particle size, mean over cloudy grid cells (cloud droplet diameter has been multiplied with 10 to fit on same scale), c –cloud ice fraction, d – height of cloud top and base, e - total water path (cloud + rain + ice), rain water path and ice water path, f – maximum updraft velocity, g – loss rate of cloud droplets due to collision-coalescence, h - total and ice precipitation rate at surface. Simulations: grey – no-emission control, green –birch pollen flux (2500 pollen/m2/s) and no rupture, blue – same birch pollen flux, SPP size from B21, orange– same birch pollen flux, SPP size from T04, brown - birch pollen flux (500 pollen/m2/s), SPP size from B21, purple - birch pollen flux (50 pollen/m2/s), SPP size from B21. Darker shades indicate simulations that include secondary ice formation (SIP). Dots – hourly average model values, lines – trend component from singular spectrum analysis with 6 hour window length.**


In this case, the cloud cover is 100% in all simulations and precipitation is distributed uniformly over the model domain. During the second half of all the simulations with SPP release, and also the one with maximum pollen emission without SPP, light precipitation is present in majority of the domain. Higher INP concentration shifts the precipitation distribution towards higher rates (Figure 10). Similarly to the liquid cloud case, the resulting precipitation is positively correlated to pollen emission (Figure 11b). Near-linear dependence is true for liquid precipitation while the solid phase levels off for higher INP

concentrations. Figure 11a shows that the precipitation resulting from 5 times lower pollen emission with the B21 SPP size distribution (P500-SPP-B21) is only ~30% lower than that of the maximum pollen emission with T04 SPP size distribution (P2500 SPP-T04). These two simulations look similar also in other variables e.g. cloud ice fraction, ice water path and maximum updraft velocity (Figure 9, brown and orange lines). Thus, depending on the assumptions about SPPs, even more commonly encountered pollen concentrations (~10 000 pollen m$^{-3}$) can have noticeable impacts on mixed phase clouds through

ice nucleation.

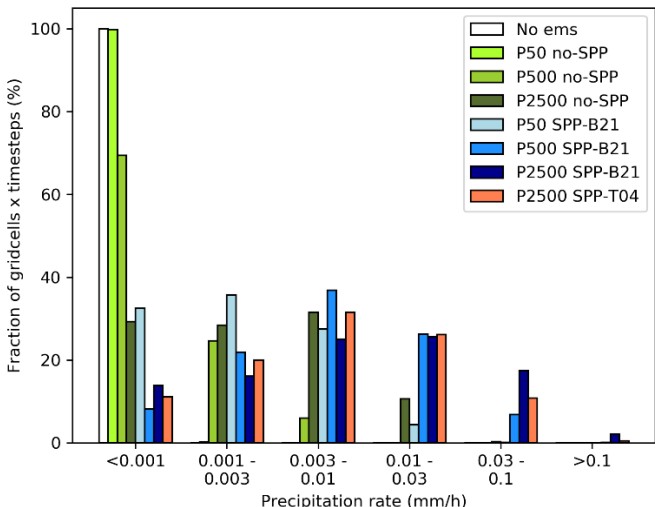

**Figure 10. Histogram of instant grid cell total precipitation (rain + ice) rates in the second half of the simulation (mm/h).**

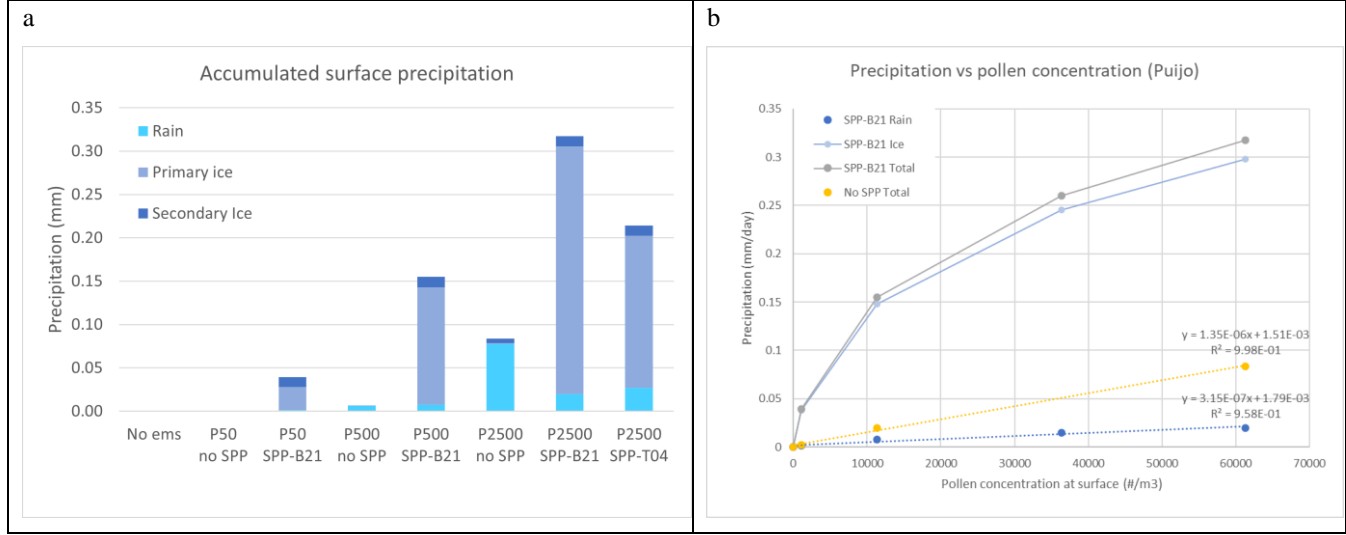

**Figure 11. a - total accumulated surface precipitation (liquid and solid) by the end of the 24-hour simulations of Puijo case with various emissions. b - Scatter plot of precipitation vs pollen concentration near surface; yellow – no rupture, the rest – SPP from B21: gray - total precipitation, dark blue – rain, light blue - ice**

Figure 12 and Figure 13 show the vertical profiles of aerosol and pollen number concentration tendencies in aerosol, cloud and rain and ice phases caused by various processes in the middle of the simulation with maximum pollen flux and rupture according to B21 size distribution.

Above the steep near-surface gradient the total pollen concentration in this case is much more uniform until the top of cloud layer (Figure 12, f), which is much lower than in the RICO case. Practically all pollens in rising airflows are activated to cloud droplets at cloud base, followed by drizzle formation during the continuing ascent (Figure 12,a-c, green lines for cloud activation and blue for rain formation through coalescence). From the f panels of Figure 12 and Figure 13 it is visible that the pollen and SPP concentrations in rain drops (darker blue line) decrease from cloud base to surface due to evaporation (orange line on Figure 12c and Figure 13c). The decrease is about one order of magnitude for the pollen particles (Figure 12f) while the concentration decreases to almost zero in the case of SPPs (Figure 13f). This means that the fraction of pollen particles deposited to the ground by rain is much larger than those of the smaller SPPs.

As seen from Figure 13, the ice nucleating SPPs are produced by pollen rupture near the cloud base (panel e, brown line) and transported to the cloud top (panel e, pink line), where the temperatures are lowest and ice nucleation rate highest (panel d, light blue line). The ice particles grow by deposition of water vapour while settling through the cloud layer. As the temperature is below zero at all vertical levels, they stay frozen until ground but do shrink due to sublimation below cloud base. The role of whole pollen grains in ice nucleation is negligible due to their low concentration and short residence time at cloud top due to high settling velocity. Instead, as seen from the dark blue line on panel d of Figure 12, they mostly end up in ice phase due to scavenging by falling ice particles. However, comparing solid and dashed light blue lines on Figure 12 and Figure 13 we

see that after falling out of the cloud the majority of the SPP-containing ice particles shrink to small sizes due to sublimation, while the pollens have accumulated more water acting as GCCN even before freezing and thus fall faster and reach ground

with larger water content, leading to increased ice-phase precipitation.

By the end of the simulation, ~12% of the emitted pollens have ruptured. The main reason for the smaller ruptured fraction than in RICO case is the shorter lifetime of pollen in air – 92% of all the emitted pollens have been deposited by the end of the simulation, while at the end of RICO case 39% were still in the air. Although this case is characterized by a strong inversion, some particles still manage to escape above the cloud layer. By the end of the 24-hour long simulation, 0.02% of the emitted

pollen and 1.0% of the produced SPPs can be found above the cloud top. As seen from panels f of Figure 12 and Figure 13, above the clouds water has evaporated from pollen and SPPs and they have deactivated and returned to aerosol phase - 70% of pollen and 87% of SPPs above the mean cloud top are found in aerosol phase.

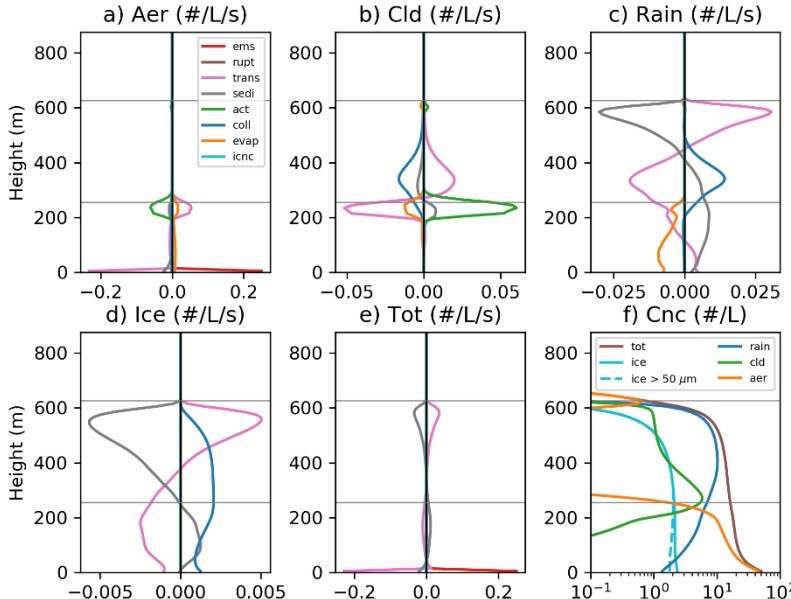

**Figure 12. Vertical profiles of pollen concentration tendencies in aerosol (A), cloud (B) and rain (C) and ice (D) phases, and the sum of those (E) due to various processes: pollen emission, rupture, vertical transport, sedimentation, cloud activation, collisions, evaporation, and ice nucleation. F – number concentration profiles of different phases. Dashed light blue line shows ice particles larger than 50 μm in diameter. Domain averaged hourly mean quantities 12 hours after the beginning of the simulations, plotted for the case with maximum pollen flux and rupture according to B21 size distribution. Grey lines denote the cloud base and top.**

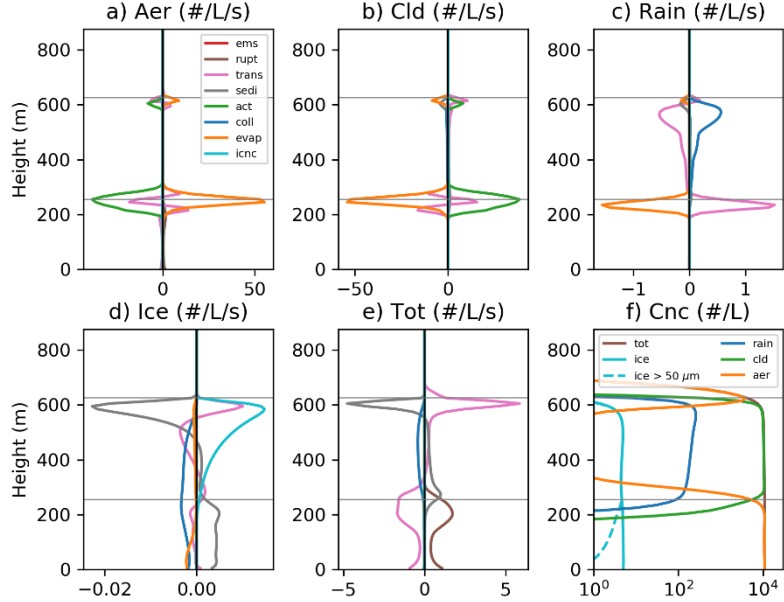


**Figure 13. A-E Vertical profiles of sub-pollen particle concentration tendencies in different phases due to various processes, F – number concentration profiles of SPPs in different phases. Panels and colors same as in Figure 12.**

## 4 Discussion and conclusions

We used the UCLALES-SALSA large eddy simulator for simulating pollen effects on precipitation in two well-described

cases, one for liquid and one for mixed-phase clouds. The simulations showed that the effects of pollens and sub-pollen particles on precipitation could be noticeable in vicinity of birch or pine forests during the most intense pollination seasons. While the giant CCN effects on liquid precipitation became noticeable only for the highest considered pollen concentrations which are rare in the observation history, the ice nucleation started influencing the clouds already at more commonly encountered pollen concentrations. However, ice nucleation was relevant only if pollen rupture was included in the simulation.

In both cases the liquid precipitation enhancement had nearly linear dependency on the emitted pollen flux, the slope of this relationship was case dependent. The ice precipitation from the mixed phase clouds levelled off for higher INP concentrations. Although a small fraction of birch pollens managed to escape the boundary layer to the free troposphere, their impact is unlikely to extend far downwind from the source regions due to dilution and deposition reducing their concentration. Pine pollens are about three times larger than birch pollen but half as dense, leading to a few times higher settling velocity and shorter

atmospheric lifetime. The fraction of pine pollens that got transported above the inversion layer by the updrafts in the convective clouds was about 5 times smaller than that of birch pollen. Thus, while more efficient as giant CCN, pine pollens are even less likely to be relevant to precipitation beyond the emission areas. However, the simulation with pine pollen led to slightly larger rain enhancement than birch pollen for same emission flux, showing that some of these extra-large particles managed to reach cloud level, and thus they could play a role in altering the precipitation for instance above the boreal forests.

The fine sub-pollen particles released when pollens rupture in humid conditions, on the other hand, have longer atmospheric lifetimes, having thus the potential to affect larger areas downwind the pollen emission. In our simulations around 1-2% of the SPPs escaped the boundary layer to free troposphere where they could accumulate and be transported for long distances to contribute to INP population further downwind. By the end of the simulation, the number of SPPs above the inversion layer was up to 4 orders of magnitude higher than that of pollen. Zhang et al. (2024) reported a similar ratio of SPPs to pollen in

upper troposphere for their modelling study when using the same number of 1000 SPPs released from a rupturing pollen grain. SPPs have two contrasting effects - while as extra CCN they slow down the precipitation formation by reducing the cloud droplet size, in mixed phase clouds they also act as efficient high-temperature ice nucleators and can lead to formation of solid precipitation. The ice nucleation by the SPPs clearly dominated over their cloud stabilizing effect, and the total precipitation was increased. Zhang et al. (2024) found that the addition of SPPs also invigorated the deep convective system they studied,

as extra latent heat was released by the enhanced cloud droplet formation. We did not observe this process in our liquid cumulus case, as we saw no significant differences in cloud top height or updraft velocities between the simulations with or without SPPs. In the mixed-phase cloud simulations we observed the opposite effect from the SPPs to the cloud dynamics than Zhang et al. (2024) - the settling of the nucleated ice particles and their subsequent vaporization below the cloud base led to a change in the vertical distribution of water vapour and a reduction of below-cloud temperature, which reduced the updraft velocities.

One potential caveat of this study is the high uncertainty of the ice nucleation parameterizations for the relatively warm temperatures of our mixed-phase cloud simulations. The ice nucleation measurements of Augustin et al. (2013) were made at temperatures below -17 °C and thus its applicability for temperatures above -10°C is uncertain. The uncertainties in birch pollen and SPPs ice nucleation efficiencies are large even for lower temperatures and have been reported to depend on the location where the pollen has been collected, the atmospheric processing it has experienced and the steps taken in preparing

the samples for the measurements (Augustin et al., 2013; Gute and Abbatt, 2018; Wieland et al., 2024). The ice nucleation rates in temperatures higher than -10 °C are too slow to measure without large uncertainty, and for this reason in majority of the cases only the median freezing temperature ($T_{50}$) is reported (Duan et al., 2023). For birch pollen, $T_{50}$ is colder than the temperatures reached in our simulations. However, this does not mean that ice nucleation would not happen at the temperatures encountered in the Puijo simulations. Recently, Wieland et al. (2024) demonstrated that birch pollen can nucleate ice in

temperatures up to at least -5.4 °C. Our ice nucleation parameterization gives similar ice nucleation rates for pollen for warmer temperatures to parameterizations that have been used in previous modelling studies. The freezing onset temperature reported for model parameterizations is usually defined as the temperature at which the ice nucleation rate exceeds a certain threshold. Hoose et al. (2010) report the freezing onset as the temperature below which the freezing rate exceeds $10^{-5}$ s$^{-1}$ and their parameterization gives the freezing onset at approximately $-8°C$ for birch pollen. Our model reaches this rate at -7.24°C, while

slower ice nucleation takes place in the model up to -2 °C for both pollen and sub-pollen particles.

    The ice nucleation parameterization of Augustin et al. (2013) used in this study was based on measurements made using pollen washing water and thus is more valid for SPPs than whole pollen grains. However, the pollen concentration at cloud top where majority of the ice nucleation happens never got high enough to make noticeable impact on ice formation even when secondary

ice formation through rime splintering was accounted for. The limited effects of pollen on cloud ice could be because of the relatively warm temperatures in the simulation. Low level clouds with much lower temperatures are unlikely to occur during strong birch pollination periods. Some trees such as hazel and alder flower earlier in spring or winter when the weather is colder (Linkosalo et al., 2017), and Gute and Abbatt (2020) report ice nucleation at temperatures higher than -10°C for alder pollen. On the other hand, the peak concentrations of alder pollen do not reach as high values as those of birch pollen. Lower temperatures can also exist if the cloud layer is higher, but in that case smaller fraction of the emitted pollen would get transported to the cloud level. This trade-off could be investigated in future studies. However, it is the unique ability of the biological particles to act as INPs at higher temperatures that can make their role in cloud dynamics important compared with the much more abundant types of INPs, such as mineral dust, that dominate the ice nucleation in colder clouds (Hoose and Möhler, 2012).

Only those of our simulation that included higher than usual pollen concentration (exceeding ~10 000 m$^{-3}$) showed a noticeable impact on clouds and precipitation. However, such pollen concentrations may be more common than reported by monitoring networks as the stations are usually designed for allergy information needs and thus located near populated areas, while high pollen concentrations are likelier to occur in forested areas far from human habitation. Additionally, while the relative effect of pollen on precipitation is likely the largest for clouds with very low precipitation like those studied here, the pollen could have larger overall impact on precipitation and cloud dynamics in different conditions than the simulated cases, for instance in clouds more prone to precipitate or with lower CCN concentrations from other sources. Sub-pollen particle impacts of similar magnitude to the maximum pollen emission cases would appear in lower pollen concentrations if more SPPs per rupturing pollen would be emitted or SPPs would be directly emitted from trees. As observational data was not available for birch pollen rupture humidity dependence and emitted SPP number, our model had to be based partly on data for other wind-pollinated plants. Using the wheat pollen rupture data probably leads to overestimation of the rupture rate for birch. More laboratory studies are required to narrow down the uncertainties in pollen rupture, and ice nucleation activity of whole pollens and SPPs.

**Code availability**

The source code of the version of UCLALES-SALSA used for the simulations can be found at https://doi.org/10.57707/FMI-B2SHARE.5B37722CC31D4B8C9EDFECA6A8DD88F6 (Prank et al., 2024)

**Data availability**

The simulation data presented in this paper is available from https://doi.org/10.57707/FMI-B2SHARE.5B37722CC31D4B8C9EDFECA6A8DD88F6 (Prank et al., 2024)

**Author contribution**

MP, SR and TR designed the study. MP performed and analysed the model simulations with assistance from JT, SR and TR. XS provided pollen observations and assisted with related literature. MP, JT, SR and TR have contributed to developing the UCLALES-SALSA model. MP prepared the manuscript with contributions from all co-authors.

**Competing interests**

The authors declare that they have no conflict of interest.

**Acknowledgements**

This work was supported by the Academy of Finland projects 322532 and 356444.

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
