# Peer review of "Can pollen affect precipitation?"

_EGUsphere, 2024_

## Author Response (AR1)

**Response to Reviewers**

We thank the reviewers for their comments and suggestions. Our answers to their comments are given below in blue.

**Reviewer #1**

**Major Comments:**

1.     The temperature range for the mixed-phase cloud used in this study is from -7 to -3 degrees Celsius, which is much higher than the temperature range of experiments in Augustin et al., 2013 (-23 to -19 degrees Celsius). The heterogeneous ice nucleation rate from Augustin et al. (2013)'s scheme decreases exponentially at warmer temperatures. Thus, the observed small effects of pollen on cloud ice in this study could be because of the relatively warmer temperatures in the simulation. Given that some studies, including Augustin et al. (2013), suggest pollen/SPP ice nucleation does not occur until lower than -10 degrees Celsius (e.g., Gute & Abbatt 2020; Matthews er al., 2023), the select cases in this study may be too warm to investigate the impact of pollen or sub-pollen particles (SPPs) as ice nucleating particles (INPs). The authors should discuss this potential limitation further in the discussion section.

It is true that the limited effects of pollen on cloud ice in this study could be because of the relatively warmer temperatures in the simulation. However, it is the unique ability of the biological particles to act as INPs at higher temperatures that can make their role in cloud dynamics important compared with the much more abundant types of INPs, such as dust, that dominate the ice nucleation in colder clouds.

The uncertainties in birch pollen and SPPs ice nucleation efficiencies are large even for lower temperatures and have been reported to depend on the location where the pollen has been collected, the atmospheric processing it has experienced and the steps taken in preparing the samples for the measurements (Augustin et al., 2013; Gute and Abbatt, 2018; Wieland et al., 2024). The ice nucleation rates in temperatures higher than -10° C are too slow to measure without large uncertainty and for this reason, in majority of the cases, only the median freezing temperature ($T_{50}$) is reported (Duan et al., 2023).  However, this does not mean that ice nucleation would not happen at higher temperatures, it just happens at slower rate. Recently, Wieland et al. (2024) demonstrated that birch pollen can nucleate ice in temperatures up to -5.4 °C.

Our ice nucleation parameterization gives ice nucleation rates for pollen for warmer temperatures that are similar to parameterizations used in previous modelling studies. The freezing onset temperature reported for model parameterizations is usually defined as the temperature at which the ice nucleation rate exceeds a certain threshold. For instance, Hoose et al. (2010) report the freezing onset as the temperature below which the freezing rate exceeds $10^{-5}$ $s^{-1}$ and their parameterization gives the freezing onset at approximately −8 °C for birch pollen. Our model reaches this rate at -7.24 °C, while slower ice nucleation takes place up to -2 °C for both pollen and sub-pollen particles.

Low level clouds with much lower temperatures are unlikely to occur during strong birch pollination periods. However, other trees such as hazel and alder flower earlier in spring or winter (Linkosalo et al., 2017), and for alder pollen, Gute and Abbatt (2020) report ice nucleation at

temperatures higher than -10 °C. On the other hand, the peak concentrations of alder pollen do not reach as high values as that of birch.

Lower temperatures can exist if the cloud layer is higher, but in that case a smaller fraction of the emitted pollen would get transported to the cloud level. This trade-off could be investigated in future studies together with alder pollen acting as a better high temperature ice nucleator.

The above paragraphs have been added to the discussion section.

2. The authors should include details about the modeling of pollen ice nucleation. It is mentioned that one SPP has one active ice site. Does this mean that regardless of the T04 or B21 size distribution, their ice nucleation activity will be the same since both distributions have the same rupture rate? If so, does the observed difference between the two experiences mainly resulting from SPP's varying impacts as CCNs? Additionally, how is the ice nucleation activity for whole pollen grains modeled? How many ice active sites does each pollen grain have? After rupture, will their ice active sites change?

Augustin et al. (2013) parameterized the average number of ice-nucleation active macromolecules in SPPs depending on their diameter. While the simplified ice nucleation rate parameterization we are using considers all the ice-nucleation active SPPs having just one active site, and thus the same ice nucleation rate, we do take into account the size dependent fraction of the finer particles not including any ice nucleating macromolecules. Thus, since the particles represented by the T04 distribution are smaller than those from B21, a smaller fraction of the 1000 emitted SPPs are ice-nucleation active. This is the main reason for the difference between these experiments. The whole and ruptured pollen grains have one ice active site each, so they have the same ice nucleation rate as the SPPs. The justification for this treatment is that it leads to a T50 value for pollen that agrees well with the literature (Gute and Abbatt, 2020) and produces freezing onset similar to Hoose et al. (2010).

We have amended the model description with the above information.

3. For the first case, Figures 2 and 4-7 show the model domain average. Given the relatively small cloud fraction over the model domain (shown in Figure 3), I would suggest to show the impacts in these figures averaged over a subset of the model domain with high cloud liquid water path. This would provide a more accurate picture of how pollen is influencing the cloud microphysical processes. (This seems less important for the second ice-cloud case, as the authors state that cloud cover is 100% in this case, line 269).

Some panels on Figure 2 in the manuscript represent the domain mean statistics, including water paths and cloud cover fraction. In-cloud aerosol and cloud droplet concentrations, cloud droplet size and cloud base and top are already averaged over cloudy grid cells. As the collision-coalescence is an in-cloud process, we have now estimated the cloudy column average by dividing the whole domain values with cloud fraction. Calculating the domain total precipitation is justified by the fact that the precipitation does not reach the surface directly below the clouds which are moving with the wind, and, as seen from Figure 1 below, it can in some cases reach the surface after the cloud that produced it has already mostly dissipated. Figures 5 and 6 in the manuscript show aerosol and rain statistics that are not all related to clouds. It is therefore not meaningful for those figures to be limited to cloudy columns.

[Figure]

*Figure 1. Instantaneous precipitation rate (contours) and cloud liquid water path (white shading) 22.5 hours after the simulation start.*

4.    The authors make a minor mention of "background aerosol" on line 244, but this should be explained in the general simulation description. Do the "no emissions" simulations include background aerosol, and if so, what is the value used?

All simulations are initialized with background aerosol size distribution and properties following the specifications in the case publications. This allows the modelling of cloud activation and further development in the absence of perturbations like pollen emissions (this is the "no emissions" simulation).

For the RICO case, VanZanten et al. (2011) specify the aerosol population as a bimodal log-normal distribution of ammonium-bisulfate with geometric mean diameters of 0.06 and 0.28 μm, geometric standard deviations of 1.28 and 1.75 and population densities of 90 and 15 cm$^{-3}$ for the first and second mode respectively.

For the Puijo case, Calderón et al. (2022) define the background aerosol as internal mixture of 12 % sulfate and 88% organic carbon, consisting of three lognormal modes with diameters of 0.039, 0.215 and 0.735 μm, geometric standard deviations of 1.5249, 1.5826 and 1.1811, and the concentrations of  274, 93 and 15 cm$^{-3}$ respectively.

We have added this information to Table 1.

5.    In general, the discussion of the model simulation results is rather sparse – some more discussion would be helpful. Specifically, more description of the process figures is needed (Figure 5-6 for the first case and 12-13 for the second case) – if there is no discussion to accompany some of the panels, maybe they number of panels should be reduced.

Discussion has been extended and more references to the relevant figures and panels have been added to the text to make it easier to connect the figures with the discussion.

**Minor comments on text:**

1. Better description of the LES simulations is needed (including acronyms). The table is not helpful and it is hard to connect the simulations to later figures (Figure 2 and onwards)

Tables were added listing all the simulations with their acronyms and parameters that were varied for each of the cases.

2. Line 55-58, the study by Zhang et al., 2024 did not use the long-time or large-scale averages.

The text was corrected to "Small number of global and regional modelling studies have investigated the impact of pollen and SPPs to precipitation, however, often the pollen concentrations in those are low, representative of long-time or large-scale averages (e.g. Werchner et al., 2022; Wozniak et al., 2018). While some studies (e.g. Zhang et al., 2024) have used realistic pollen emissions, their emissions represent an average pollen year, while during intense flowering on mast years the concentrations can locally reach many times what is used in those studies."

3. What percentage of whole pollen grains rupture to produce SPPs during the simulation?

By the end of the RICO simulation, ~23% of the emitted pollens have ruptured. For Puijo this number is smaller (~12%). The main reason for the smaller ruptured fraction in Puijo case is the shorter lifetime of pollen in air – 92% of all the pollens emitted have been deposited by the end of the simulation, while at the end of RICO case 39% are still in the air.

4. Line 124 – state where/what time of year the RICO campaign is

RICO case was selected as a simple prototype for precipitating shallow cumulus convection. The field campaign took place over the Northwestern Atlantic in winter. This information was added to the description of the model simulations

5. Line 163 – "significant" – was this based on statistical significance testing?

It is hard to assess the statistical significance of the difference between single timeseries, while the LES simulations are computationally too expensive to run for obtaining sufficient number of replicates. However, the timeseries of each simulation has a trend and variations with magnitudes sometimes exceeding the difference between the trends. In Figure 2 below the timeseries have been smoothed using singular spectrum analysis with 6-hour window length to make them easier to distinguish from each other and hourly averaged model output is plotted as dots to visualize the model variability. We have repeated one of the simulations with different random perturbations for turbulence development (red and blue lines). The difference between these two simulations can be used to visually estimate which case-to-case differences exceed the level of model noise. For instance, the RICO simulations with and without pollen (green and grey lines on Figure 2) do not differ beyond the level of model noise for cloud cover fraction (panel D), cloud top and base heights (panel E), and maximum updraft velocity (panel I).To clarify that we are not using statistical testing, we have changed word "significant" to "noticeable". We have also redrawn the timeseries figures including smoothed lines to make the differences between the noisy simulations better visible.

[Figure]

*Figure 2. Timeseries of model variables for the RICO case. Dots – hourly average model values, lines – trend component from singular spectrum analysis with 6 hour window length.*

6.    Line 185-6: "The effect is larger in the case of the B21 parameterization..." – I don't see this in Figure ... can you clarify?

The sentence has been corrected to: "while the cloud droplets are slightly smaller for the B21 parameterization (Figure 2b), it does not seem to influence the total precipitation in the highest emission case (Figure 4a), for which both of the parameterizations were tested."

7.    Line 248: "As seen from Figures 9-11..." – it's very difficult for a reader to assess this conclusion without the description of the individual figures. I would suggest to discuss the figures one by one before making such a statement

The text has been clarified and references to specific figures and panels added.

8.    Line 289: "... the fraction of them reaching the ground is much larger than those with smaller core particles." How can the authors make this conclusion about the size from the figure (which isn't included?).  Perhaps I am missing this point here.

The text was changed as follows:

Comparing the f panels of Figures 12 and 13 shows that the pollen and SPP concentrations in rain drops (darker blue line) decrease from cloud base to surface due to evaporation (orange line on the c panels of Figs 12 and 13). The decrease is about one order of magnitude for the pollen particles (Fig. 12f) while the concentration decreases to almost zero in the case of SPPs (Fig. 13f). This means that the fraction of pollen particles deposited to the ground by rain is much larger than those of the smaller SPPs.

9.    Line 319: "trice" – do the authors mean 3x? The use of "trice" seems like a typo – while technically  it is a word it is not in common usage.

"Trice" was changed to "3 times"

10.   The final section of the manuscript could frame the caveats more clearly in context with the conclusions. As written, it feels like a laundry list of items without a clear path forward on next steps.

The final section has been largely rewritten.

**Minor comments on tables and figures:**

1.    Table 1 – use superscripts; is the conversion from flux to concentration based on modeled values?

Yes, preliminary model runs were used to calibrate the fluxes. This information has been added to the methods section and the table has been fixed.

2.    Figure 3 – What is the value of the white color, as this is not included on the color bar? Include the instantaneous time displayed in panel A in the caption.  Additionally, axes aren't labeled.

In the white areas the total liquid water path exceeds 1000 g/m2. The color scale has been redrawn to clarify that. The x and y axes were re-labeled as south-north and west-east

displacement from model origin. The time was added to figure caption. Additionally, instead of the total water path (rain + cloud), the figure now shows cloud water path.

3.      4a, "blue – no rupture", do you mean green- no rupture?

Yes. The figure caption was corrected.

4.      6a and Fig. 13a, why does the SPP concentration tendency in aerosol phase show much larger values from evaporation/transportation than from rupture?

Pollen rupture in the model is a relatively slow process, with a time scale of 2.5 hours at highest humidities. Activation to cloud droplets at the cloud base, on the other hand, is a fast process that involves all the SPPs produced cumulatively during the simulation and is balanced by evaporation back to aerosol phase after leaving the cloud. Transport includes both grid scale and sub-grid scale vertical transport, and as domain average it can be considered as mixing that acts to reduce the vertical gradients produced by the other processes. The processes that move particles from one phase to the other are not visible on panel D of Figure 6 and panel E of Figure 13 that depict the sum of all the phases, so there the rupture and following transport are the dominating processes. For SPPs sedimentation is too slow to be visible.

5.      Figure 7 – how are the layer fluxes calculated from the model output?

UCLALES-SALSA output includes the vertical profiles of concentration tendencies due to various processes, and the height of the inversion layer. To obtain the net flux through the inversion layer, the tendencies due to advection, sub-grid mixing, and sedimentation are integrated over the model levels above the boundary layer.

6.      Figures 3 & 9 – the many lines (solid/dashed + colors) make it very hard to understand this figure and parse through the lengthy caption. This needs to be more clear in the legends.  Perhaps including the solid/dashed legend on the individual panels would make this more accessible to the reader.

The figures have been redrawn with smoothed lines to improve the readability and solid/dashed has been added to panel titles.

7.      Figure 9d: is the ice fraction really 0.01-0.03%? That is extremely low and likely within the model noise. Or do the authors mean 1-3%?

The fraction given in the manuscript is not an error and, in reality, it is not so low. Also, while the ice particles do indeed make up a very small fraction of the total cloud droplets, their effects are well beyond model noise.  As seen from Figure 13 in the manuscript, panel f, the ice number concentration in the cloud is about 5 particles per liter, which is well within the ice crystal concentrations range observed by Gryspeerdt et al. (2018) and increases further towards the end of the simulation. One reason for the ice fraction to seem low is the relatively higher background aerosol and cloud droplet number in this case compared to e.g. clean marine atmosphere. While the rupture is a relatively slow process and the ice nucleation is also very limited due to low IN efficiency in those temperatures, the SPP concentration is high and thus the resulting ice number concentration is noticeable. Once formed, they grow fast and end up being much larger than the cloud droplets, so by the end of the simulation with maximum ice formation (P2500 SPP-B SIP) about half of the total (liquid + ice) water path is actually frozen. Due to the settling of the ice particles the liquid+ice water path also reduces by half and a drop is visible in the cloud height by the end of the simulation.

8. 13f, why is the concentration of SPP consistent between 200 to 600 m?

It is not exactly constant, though it looks like that on the figure due to the logarithmic x axis. A logarithmic axis is used so that the very low concentrations of rain drops and ice would be visible. The reason for small gradients is most probably the efficient mixing in this layer.

**Reviewer #2**

1. The INP from pollen that is formed in Mixed-phase condition case is between 0 to -6 C while the parameterization that is used for pollen INP calculation (Augustin et al., 2013) is for temperatures below -17C

The uncertainties in birch pollen and SPPs ice nucleation efficiencies are large even for lower temperatures and have been reported to depend on the location where the pollen has been collected, the atmospheric processing it has experienced and the steps taken in preparing the samples for the measurements (Augustin et al., 2013; Gute and Abbatt, 2018; Wieland et al., 2024). The ice nucleation rates in temperatures higher than -10 °C are too slow to measure without large uncertainty and for this reason in majority of the cases only the median freezing temperature ($T_{50}$) is reported (Duan et al., 2023). However, this does not mean that ice nucleation would not happen at higher temperatures, it just happens at slower rate. Recently, Wieland et al. (2024) demonstrated that birch pollen can nucleate ice in temperatures up to -5.4 °C.

The parameterization we use agrees reasonably well with those used in other modelling studies. The freezing onset temperature reported for model parameterizations is usually defined as the temperature at which the ice nucleation rate exceeds a certain threshold. For instance, Hoose et al. (2010) report the freezing onset as the temperature below which the freezing rate exceeds $10^{-5}$ $s^{-1}$ and their parameterization gives the freezing onset at approximately $-8$ °C for birch pollen. Our model reaches this rate at -7.24 °C, while slower ice nucleation is allowed to take place up to -2 °C for both pollen and sub-pollen particles.

2. There is no discussion about other types of aerosols that contribute to cloud formation and affect precipitation rates. (dust )

The following paragraphs were added to the Introduction section and the description of background aerosol was added to the Methods section.

Typically cloud droplets are formed on hygroscopic aerosol particles (composing of sulfate, nitrate, sea salt, organic aerosol, etc.), with the number of cloud condensation nuclei (CCN) in cubic centimeter ranging from below 100 in clean marine atmosphere to thousands in polluted areas (Seinfeld and Pandis, 1998). Thus, even the highest observed pollen concentrations are too low to noticeably influence the CCN concentration.

As shown by (Houghton, 1938), rain drops in liquid clouds are formed by collisions of cloud droplets of different sizes, and presence of a small number of large hygroscopic CCN is an essential factor for the appearance of cloud droplets of different size in the same location. Coarse sea spray and mineral dust have been shown to act as such giant CCN (GCCN), enhancing the collision-coalescence rate and starting the development of drizzle (Adebiyi et al., 2023; Feingold et al., 1999). (Feingold et al., 1999) found a noticeable enhancement in drizzle formation from GCCN concentrations as low as 0.001 cm$^{-3}$, which is well in the range of observed pollen concentrations.

At temperatures warmer than the homogeneous freezing limit at ~-38 °C, ice in the clouds is formed heterogeneously on particles which can initiate freezing. Depending on temperature, different solid particles can act as ice nucleating particles. In colder temperatures (below -15 °C) ice nucleation is dominated by dust, while primary biological aerosol particles are the most efficient INPs for temperatures warmer than -10 °C (Hoose and Möhler, 2012).

The background aerosol in the UCLALES-SALSA simulations is identical to the original publications of the cases and stays the same between the simulations with different pollen fluxes and SPP production.

3.     The size of the pollen can be improved using a more recent approximation from Hoose et 2010.

Hoose et al. (2010) are using 30 μm as pollen diameter and simulate a single pollen species representing the pollen of all plant species, while we are simulating specifically the pollen of birch and pine with their measured sizes. Our birch pollen size of 22 μm is the same as used for birch pollen in other models e.g. Siljamo et al. (2013).

4.     There is no discussion about the simulated updraft velocity which controls all meteorological variables .

The timeseries of domain maximum updraft velocities were added to Figures 2 and 9 of the manuscript and are now discussed in the Results section.

For the RICO warm cumulus case, the updraft velocity does not change between the different simulations. For the mixed phase case, lower updraft velocities are seen for the simulations with pollen and SPPs in the second half of the simulations. The reason for this is the change in the vertical profiles of water caused by the settling of the ice particles. While the number concentration of the ice particles is small compared to the liquid cloud droplets, the ice particles are much larger, and by the end of the simulation with maximum INP production (P2500 SPP-B SIP) about half of the water in cloud is frozen (see Figure 4e). Notable amount of water is removed from the cloud layer by settling of the ice particles which evaporate below cloud base and this changes the humidity profile. In the no-emission base case, the relative humidity at surface falls to ~80% by the end of the simulation (

Figure 3b). For the maximum ice nucleation case it stays around 90% and a small drop compared with the base case is also seen in the near-surface temperature due to the energy spent for sublimation (

Figure 3d,e). This results in less buoyancy and slower updraft velocities.

[Figure]

*Figure 3. Profiles of domain minimum temperature, mean relative humidity and maximum updraft velocity for the no-emission control (upper row) and the maximum INP simulation (P2500 SPP-B21 SIP, lower row) (initial, 6, 12, 18 and 24 hours after simulation start, from darker to lighter lines).*

[Figure]

*Figure 4. Timeseries of the Puijo simulations. Panels: a – in-cloud interstitial aerosol and cloud droplet concentration, b – cloud droplet and ice particle size, mean over cloudy grid cells (cloud droplet diameter has been multiplied with 10 to fit on same scale), c – cloud ice fraction, d – height of cloud top and base, e - total water path (cloud + rain + ice), rain water path and ice water path, f – maximum updraft velocity, g – loss rate of cloud droplets due to collision-coalescence, h - total and ice precipitation rate at surface. Simulations: grey – no-emission control, green – birch pollen flux (2500 pollen/m2/s) and no rupture, blue – same birch pollen flux, SPP size from B21, orange – same birch pollen flux, SPP size from T04. Darker shades indicate simulations that include secondary ice formation (SIP). Dots – hourly mean model data, lines – trend component from singular spectrum analysis with 6 hour window.*

5.    It is not clear if pollen is emitted insoluble and turn over to soluble. However they set the hygroscopicity parameter to 0.16 for both SPPs and whole pollens.

Pollen is always treated as a soluble and the hygroscopicity parameter stays the same throughout the simulation. This is in the range reported in the literature, e.g. Griffiths et al. (2012) reported pollen hygroscopicities in the range of 0.05 to 0.22 and found that the wettability and large size of pollen grains leads to them activating to cloud droplets in supersaturations of 0.0015% and lower. Clarifications were added to the Methods section.

6.    ``` as particles with diameters in the range of tens of micrometres activate easily as cloud droplets as long as they are not hydrophobic, this approximation should have limited impact.``` this is contrary with model's results that found SPPs and pollen above cloud top. Fig 12-13

This is due to the evaporation of cloud droplets at cloud edges - while all the pollens activate to cloud droplets at the cloud base, they will deactivate in the dry air above or at the edges of the clouds if they happen to escape the cloud.

**Minor comments**

Line 40: After INPs add a reference

Done

Line 118: check Celsius acronym

Done

Line 160: theme font is not everywhere the same

The font of the text has been checked

Line 267: '' (2500 pollen/m2/s), no rupture'' to '' (2500 pollen/m2/s) and no rupture''

Done

Line 273: Figure 11, A not right

"Right" was removed, as the sentence references the whole figure

Figure 5, ABCD should be capital letters

The journal guidelines specify the usage of lower-case letters.

Figure 5 d should be deleted. the sum gives the budget ?

Yes, it's the budget and we prefer to keep it for completeness' sake.

Figure 8 y axis has no units

Units have been added

**References**

Adebiyi, A., Kok, J. F., Murray, B. J., Ryder, C. L., Stuut, J. B. W., Kahn, R. A., Knippertz, P., Formenti, P., Mahowald, N. M., Pérez García-Pando, C., Klose, M., Ansmann, A., Samset, B. H., Ito, A., Balkanski, Y., Di Biagio, C., Romanias, M. N., Huang, Y. and Meng, J.: A review of coarse mineral dust in the Earth system, Aeolian Res., 60(November 2022), doi:10.1016/j.aeolia.2022.100849, 2023.

Augustin, S., Wex, H., Niedermeier, D., Pummer, B., Grothe, H., Hartmann, S., Tomsche, L., Clauss, T., Voigtländer, J., Ignatius, K. and Stratmann, F.: Immersion freezing of birch pollen washing water, Atmos. Chem. Phys., 13(21), 10989–11003, doi:10.5194/acp-13-10989-2013, 2013.

Calderón, S. M., Tonttila, J., Buchholz, A., Joutsensaari, J., Komppula, M., Leskinen, A., Hao, L., Moisseev, D., Pullinen, I., Tiitta, P., Xu, J., Virtanen, A., Kokkola, H. and Romakkaniemi, S.: Aerosol-stratocumulus interactions : Towards a better process understanding using closures between observations and large eddy simulations, Atmos. Chem. Phys. Discuss. [preprint], in review, doi:https://doi.org/10.5194/acp-2022-273, 2022.

Duan, P., Hu, W., Wu, Z., Bi, K., Zhu, J. and Fu, P.: Ice nucleation activity of airborne pollen: A short review of results from laboratory experiments, Atmos. Res., 285, 106659, doi:10.1016/j.atmosres.2023.106659, 2023.

Feingold, G., Cotton, W. R., Kreidenweis, S. M. and Davis, J. T.: The impact of giant cloud condensation nuclei on drizzle formation in stratocumulus: Implications for cloud radiative properties, J. Atmos. Sci., 56(24), 4100–4117, doi:10.1175/1520-0469(1999)056<4100:TIOGCC>2.0.CO;2, 1999.

Griffiths, P. T., Borlace, J. S., Gallimore, P. J., Kalberer, M., Herzog, M. and Pope, F. D.: Hygroscopic growth and cloud activation of pollen: A laboratory and modelling study, Atmos. Sci. Lett., 13(4), 289–295, doi:10.1002/asl.397, 2012.

Gryspeerdt, E., Sourdeval, O., Quaas, J., Delanoë, J., Krämer, M. and Kühne, P.: Ice crystal number concentration estimates from lidar-radar satellite remote sensing - Part 2: Controls on the ice crystal number concentration, Atmos. Chem. Phys., 18(19), 14351–14370, doi:10.5194/acp-18-14351-2018, 2018.

Gute, E. and Abbatt, J. P. D.: Oxidative Processing Lowers the Ice Nucleation Activity of Birch and Alder Pollen, Geophys. Res. Lett., 45(3), 1647–1653, doi:10.1002/2017GL076357, 2018.

Gute, E. and Abbatt, J. P. D.: Ice nucleating behavior of different tree pollen in the immersion mode, Atmos. Environ., 231(April), 117488, doi:10.1016/j.atmosenv.2020.117488, 2020.

Hoose, C. and Möhler, O.: Heterogeneous ice nucleation on atmospheric aerosols: A review of results from laboratory experiments., 2012.

Hoose, C., Kristjánsson, J. E., Chen, J. P. and Hazra, A.: A classical-theory-based parameterization of heterogeneous ice nucleation by mineral dust, soot, and biological particles in a global climate model, J. Atmos. Sci., 67(8), 2483–2503, doi:10.1175/2010JAS3425.1, 2010.

Houghton, H. G.: Problems Connected with the Condensation and Precipitation Processes in the Atmosphere *, Bull. Am. Meteorol. Soc., 19(4), 152–159, doi:10.1175/1520-0477-19.4.152, 1938.

Linkosalo, T., Le Tortorec, E., Prank, M., Pessi, A. M. and Saarto, A.: Alder pollen in Finland

ripens after a short exposure to warm days in early spring, showing biennial variation in the onset of pollen ripening, Agric. For. Meteorol., 247, 408–413, doi:10.1016/j.agrformet.2017.08.030, 2017.

Seinfeld, J. H. and Pandis, S. N.: Atmospheric chemistry and physics: from air pollution to climate change, Wiley-Interscience, New York. [online] Available from: http://books.google.com/books?id=Z1idQgAACAAJ, 1998.

Siljamo, P., Sofiev, M., Filatova, E., Grewling, L., Jäger, S., Khoreva, E., Linkosalo, T., Ortega Jimenez, S., Ranta, H., Rantio-Lehtimäki, A., Svetlov, A., Veriankaite, L., Yakovleva, E. and Kukkonen, J.: A numerical model of birch pollen emission and dispersion in the atmosphere. Model evaluation and sensitivity analysis, Int. J. Biometeorol., 57, 125–136, doi:10.1007/s00484-012-0539-5, 2013.

VanZanten, M. C., Stevens, B., Nuijens, L., Siebesma, A. P., Ackerman, A. S., Burnet, F., Cheng, A., Couvreux, F., Jiang, H., Khairoutdinov, M., Kogan, Y., Lewellen, D. C., Mechem, D., Nakamura, K., Noda, A., Shipway, B. J., Slawinska, J., Wang, S. and Wyszogrodzki, A.: Controls on precipitation and cloudiness in simulations of trade-wind cumulus as observed during RICO, J. Adv. Model. Earth Syst., 3(2), doi:10.1029/2011MS000056, 2011.

Werchner, S., Gute, E., Hoose, C., Kottmeier, C., Pauling, A., Vogel, H. and Vogel, B.: When Do Subpollen Particles Become Relevant for Ice Nucleation Processes in Clouds?, J. Geophys. Res. Atmos., 127(24), 1–14, doi:10.1029/2021JD036340, 2022.

Wieland, F., Bothen, N., Schwidetzky, R., Seifried, T. M., Bieber, P., Pöschl, U., Meister, K., Bonn, M., Fröhlich-Nowoisky, J. and Grothe, H.: Aggregation of ice-nucleating macromolecules from Betula pendula pollen determines ice nucleation efficiency (preprint), EGUSphere, (April), 1–19 [online] Available from: https://doi.org/10.5194/egusphere-2024-752, 2024.

Wozniak, M. C., Solmon, F. and Steiner, A. L.: Pollen Rupture and Its Impact on Precipitation in Clean Continental Conditions, Geophys. Res. Lett., 45(14), 7156–7164, doi:10.1029/2018GL077692, 2018.

Zhang, Y., Subba, T., Matthews, B. H., Pettersen, C., Brooks, S. D. and Steiner, A. L.: Effects of pollen on hydrometeors and precipitation in a convective system, JGR Atmos., 129, e2023JD039891, doi:10.1029/2023JD039891, 2024.

---

## Author Response (AR2)

**Response to Editor**

We would like to express our gratitude to the editor for handling our manuscript. The requested technical corrections have been implemented.

Line 399 'Figure 2f' I think this should be 'Figure 2h'.

Thanks for noticing the mistake, the reference has been corrected.

Line 462 Figure 11 comes before Figure 10 that is cited in line 502. Please order figures the way they appear in the text.

The order of figures 10 and 11 was swapped.

Captions of Figure 2 and Figure 9, please indicate to which case each of them refers (liquid cumulus, mixed phase).

Figure captions were amended.